**Data Availability Statement:** The data supporting this paper have been uploaded to the Gene Expression Omnibus as GSE178435.

**Funding:** This work was funded by the Sanford Children's Health Research Center and Brightseed,

# A potent HNF4α agonist reveals that HNF4α controls genes important in inflammatory bowel disease and Paneth cells

**Seung-Hee Lee, Vimal Veeriah, Fred Levine***

SBP Medical Discovery Institute, La Jolla, CA, United States of America

* flevine@sbpdiscovery.org

## Abstract

HNF4α has been implicated in IBD through a number of genome-wide association studies. Recently, we developed potent HNF4α agonists, including N-trans caffeoyltyramine (NCT). NCT was identified by structural similarity to previously the previously identified but weak HNF4α agonists alverine and benfluorex. Here, we administered NCT to mice fed a high fat diet, with the goal of studying the role of HNF4α in obesity-related diseases. Intestines from NCT-treated mice were examined by RNA-seq to determine the role of HNF4α in that organ. Surprisingly, the major classes of genes altered by HNF4α were involved in IBD and Paneth cell biology. Multiple genes downregulated in IBD were induced by NCT. Paneth cells identified by lysozyme expression were reduced in high fat fed mice. NCT reversed the effect of high fat diet on Paneth cells, with multiple markers being induced, including a number of defensins, which are critical for Paneth cell function and intestinal barrier integrity. NCT upregulated genes that play important role in IBD and that are downregulated in that disease. It reversed the loss of Paneth cell markers that occurred in high fat diet fed mice. These data suggest that HNF4α could be a therapeutic target for IBD and that the agonists that we have identified could be candidate therapeutics.

## Introduction

HNF4α is a nuclear receptor transcription factor that is expressed predominantly in the liver, intestine, pancreas, and kidney. In the liver, where it is best studied, it plays important an important role in metabolic homeostasis, including gluconeogenesis the urea cycle, and lipid metabolism [1–4]. However, its role in other organs where it is expressed, including the pancreatic islet [5, 6], and the kidney [7–9], its role varies considerably.

In the intestine, genetic deletion of HNF4α leads to loss of mucin-associated genes, increased intestinal permeability, loss of intestinal stem cell renewal (PMID: 31759926) and predisposes to inflammatory bowel disease [10] as well as loss of brush border genes [11]. In humans, HNF4α mRNA was decreased in intestinal biopsies from patients with inflammatory bowel disease (IBD) and HNF4α has been linked to IBD in multiple GWAS studies [12–14].

Because of its central role in pathophysiologic processes that affect multiple organs, attempts have been made to discover HNF4α ligands [15], but despite initial claims of success

Inc. The funders had no role in study design, data collection and analysis, decision to publish, or preparation of the manuscript.

**Competing interests:** I have read the journal's policy and the authors of this manuscript have the following competing interests: FL holds equity in Brightseed.

[16, 17], those were ultimately unsuccessful, being irreproducible by us and others [18, 19]. In terms of natural ligands, a subset of fatty acids were known to be bound in the HNF4α ligand binding pocket [20], they were thought to play a structural rather than regulatory role, because the bound fatty acids were not exchangeable in the context of the ligand binding domain constructs that were typically used for ligand screening [15, 20]. More recently, linoleic acid was shown to bind to HNF4α and to be exchangeable in vivo [21].

In the process of a project to find modulators of the human insulin promoter, with the goal of developing therapeutics for metabolic disease, we developed a novel cell-based assay in which GFP is expressed under the control of the human insulin promoter [22]. We used that assay in multiple high-throughput phenotypic screens [18, 22–24]. The initial hit from screening the assay was a synthetic compound that we ultimately finding to be a potent antagonist of HNF4α [18]. Once an antagonist had been identified and we knew that the assay was sensitive to HNF4α activity, we screened for agonists, initially focusing on known drugs that could reverse the repressive effect of fatty acids on insulin promoter activity, with the goal of finding compounds that might be relevant to metabolic syndrome and type 2 diabetes. That study found that the known drugs alverine and benfluorex, which are structurally similar, are HNF4α agonists [23]. Benfluorex was used to treat type 2 diabetes until its withdrawal because of side effects [25] but did not have a clearly established mode of action [26]. Of note, alverine is used in the treatment of irritable bowel syndrome [27], which has some similarities to IBD [28]. Unfortunately, but not unexpectedly, the initial agonists were weak and had poor PK [23], which led us to seek improved HNF4α agonists.

Recently, by screening compounds with structures similar to alverine and benfluorex, we found that N-trans caffeoyltyramine (NCT) is a much more potent activator of HNF4α [29]. It interacts directly and so is a true agonist [29] and exhibits specificity for HNF4α, as HNF4α siRNA ablated its effect [29]. Our initial studies with NCT administration employed intraperitoneal (IP) injection into diet-induced obese mice that had severe hepatic steatosis. Encouragingly, NCT reversed nonalcoholic fatty liver disease (NAFLD). The mechanism involved induction of lipophagy through a pathway that involved the regulation of dihydroceramide production [29]. Having demonstrated the therapeutic potential of the novel HNF4α activator NCT in the liver, it was compelling to study its use in other diseases in which HNF4α is known to play an important role. To that end, we studied the effect of NCT on gene expression in the intestine.

## Materials and methods

### Poly-A fragment sequencing

PolyA RNA was isolated using the NEBNext® Poly(A) mRNA Magnetic Isolation Module and barcoded libraries were made using the NEBNext® Ultra II™ Directional RNA Library Prep Kit for Illumina®(NEB, Ipswich MA). Libraries were pooled and single end sequenced (1X75) on the Illumina NextSeq 500 using the High output V2 kit (Illumina Inc., San Diego CA).

### Bioinformatics

For analysis of RNA-seq data from control and NCT-treated mice, read data was processed in BaseSpace (basespace.illumina.com). Reads were aligned to Mus musculus genome (mm10) using STAR aligner (https://code.google.com/p/rna-star/) with default settings [30]. Gene expression estimation was performed using Cufflinks version 2.2.1 [31] and differential transcript expression was determined using DESeq2 (https://bioconductor.org/packages/release/bioc/html/DESeq2.html) [32]. Genes that were flagged as "significant" in DESeq2 had a false

discovery rate (FDR) corrected P-value <0.05. A fold change cutoff of 2-fold was then applied to identify significantly altered genes.

## STRING network analysis

STRING (https://string-db.org) shows protein-protein interaction networks. The top 122 genes upregulated by >2.9 fold by NCT in HFD+NCT treated mouse intestine (GSE178435) were analyzed. STRING functional enrichment analysis was also performed with same gene list.

## RT-PCR

Total RNA was isolated from small intestine tissues using Trizol (Invitrogen). cDNA was amplified using 3mg of total RNA using qScript cDNA SuperMix (Quanta BioSciences, Beverly, MA, USA). Quantitative real time PCR (RT-PCR) analysis was performed using SYBR® Select Master Mix (Applied Biosystems) and an ABI 7900HT thermal cycler (Applied Biosystems, Thermo Fisher Scientific) using the primers in Table 1. Ct values were normalized to 18s rRNA and are expressed as fold change over samples from mice fed normal chow.

## Mice

12-week-old C57BL/6J DIO male mice (cat#380050) and age matched C57BL/6J with normal chow diet mice were purchased from Jackson Laboratory and were fed with high fat diet containing 60 kcal% fat (Research Diets cat #D12492) or normal chow. Mice were maintained in a 12-hour light/day cycle. After 2 weeks of acclimation, mice with similar body weights were randomly assigned to HFD+DMSO (high fat diet with DMSO) control or treatment HFD +NCT (high fat diet with NCT). To test the effect of NCT (Sundia MediTech Company, Ltd., Custom synthesis), 200 mg/kg was injected IP bid for 14 days. Ten cm of small intestine proximal to the duodenum was dissected, washed in cold PBS, cut into pieces and distributed for

**Table 1. RT-PCR primer sequences (m- mouse, F- forward primer, R- reverse primer).**

| Gene | Primer sequence | Gene | Primer sequence |
|---|---|---|---|
| mMap3k6 F | ATG TTC GTG TTG GAC TCG CT | mDuox2 F | CTG GGC TTG TTG TGG TTT CG |
| mMap3k6 R | GGC ACT CAC GTT CCT TCT CA | mDuox2 R | AGC CTG GCT ATA ACT GGG GA |
| mMylip F | GGG AGC AAA GGT GAG AGC TT | mTrpm6 F | GCG CTC CGT TTG TCA AGT TT |
| mMylip R | GCT CCT TAT GCT TCG CAA CG | mTrpm6 R | GTC AGG AAA GAA CCC GGA GG |
| mNos2 F | TGA GGC TGA AAT CCC AGC AG | mDdah1 F | GCT CAA AGG GAG CAT GGA GT |
| mNos2 R | AGG CCT CCA ATC TCT GCC TA | mDdah1 R | CCT TGT GAT TAG GGC CGT GT |
| mMgat4c F | GCA GAA GCC AGA AGA GGG TT | mSlc34a2 F | AGA GGA GGA GAA GGA GCA GG |
| mMgat4c R | TAC AGC ATG GGA ACG TGC TT | mSlc34a2 R | CAC TGT TTG GAC TTG GCT GC |
| mPlb1 F | GTT CCG CAA ACG CTT TCC TT | mAdcy8 F | CCT GGG GGA CTG CTA CTA CT |
| mPlb1 R | GGG CTC TGG GTA CCT CCT TA | mAdcy8 R | CTC CCA GGG ATT CCT CCA GA |
| mDefa22 F | GAC CAG GCT GTG TCT GTC TC | mEnpp7 F | CAA CCC CAG GAT CAC ACC TC |
| mDefa22 R | GCC TCA GAG CTG ATG GTT GT | mEnpp7 R | AGG GAT CTG ATG GCC TGT CT |
| mPlb1 F | GTT CCG CAA ACG CTT TCC TT | mXpnpep2 F | CCC TTG ATC TAC TCG TCG CC |
| mPlb1 R | GGG CTC TGG GTA CCT CCT TA | mXpnpep2 R | CGA CTA TCG GTC CAG ACA GC |
| mSlc10a2 F | ATG TGG GTT GAC TCG GGA AC | mHnf4a R | ATG TAC TTG GCC CAC TCG AC |
| mSlc10a2 R | GGG GGA GAA GGA GAG CTG TA | mHnf4a F | GGT CAA GCT ACG AGG ACA GC |
| mSlc5a8 F | TTT TTG TGG CCT GCG CTT AC | m18s R | TTC GAA TGG GTC GTC GCC GC |
| mSlc5a8 R | AGC CAT AGG TTT CAA GGG GC | m18s F | ACC AAC CCG GTG AGC TCC CT |

 

analyses. For RNA isolation, samples were snap frozen using liquid nitrogen and stored at -80°C. For immunofluorescence analysis, samples were fixed in 4% of cold PFA (Santa Cruz Bio) and processed for histology. Mice were anesthetized with Avertin (IP, 0.015–0.017 ml/g) and the depth of anesthesia assessed by either ear or toe pinch before treatment. Mice were observed closely throughout treatment for signs of distress such as labored breathing, change of skin color and for signs of consciousness. Once mice were awake and mobile, they were monitored twice daily. Mice were housed in isolated cages in ventilated racks. Mice were euthanized with CO2 after administering anesthesia. All animal experiments were approved by the Institutional Animal Care and Use Committee (IACUC) of the Sanford Burnham Prebys Medical Discovery Institute in accordance with national regulations.

### Immunofluorescence and analysis

Frozen intestine sections were permeabilized using 0.3% Triton-X and incubated in antigen retrieval solution (Antigen retrieval citrate, Biogenex) at sub boiling temperature for 10min. Subsequently, sections were incubated with blocking buffer containing 5% normal donkey serum (Jackson Immuno Research) followed by incubation overnight at 4°C with primary antibody against HNF4α (1:800, Cat# PP-H1415-00, R&D Systems) or Lysozyme (1:200, Cat# PA1-29680, Invitrogen). Sections were washed and incubated with anti-mouse or rabbit secondary antibody coupled with Alexa flour 488 (1:400, Invitrogen) or with Rhodamine Red (1:400, Jackson Immuno) for 1 hour at room temperature and counterstained with DAPI (40,6-diamidino-2- phenylindole, Sigma Aldrich). Slides were mounted using fluorescence mounting medium and images were obtained at 40x magnification using an Olympus IX71 fluorescence microscope. Fluorescence intensity of HNF4α-stained nuclei was calculated using MetaMorph TL software (version 7.6.5.0, Olympus).

### Statistical analysis

Data are presented as mean ± SEM of three or more samples as indicated. Statistical significance was assessed using Student's *t*-test or ANOVA.

## Results

### High fat diet represses intestinal HNF4α and that is reversed by NCT

To study the effect of HNF4α activation by NCT on the intestine, we isolated small intestine from obese mice treated by IP injection with NCT or DMSO [29]. The intestine was processed for immunohistochemistry and RNA isolation. HNF4α was strongly decreased in the intestine of DIO mice (Fig 1), consistent with our finding that fatty acids, present at high levels in obese mice, act as HNF4α antagonists [18]. NCT reversed the effect of high fat diet on HNF4α expression in the intestine, inducing a large increase that extended throughout the length of the intestinal villus, from crypt to villus tip. In fact, the level of intestinal HNF4α expression was higher in DIO mice treated with NCT than in mice on control diet as determined by immunostaining (Fig 1B) and by RT-PCR (Fig 1C).

### Analysis of genes affected by NCT

Having shown that NCT increased HNF4α expression in the intestine, we performed RNA-seq on RNA isolated from the intestine of DIO mice administered NCT or DMSO (GSE178435). Analysis of genes that were identified by DESeq2 [33] as being significantly altered by NCT revealed three classes: obesity-associated, IBD-associated (Table 2), and Paneth cell associated (Table 3). STRING network and enrichment analysis of the RNA-seq data

 

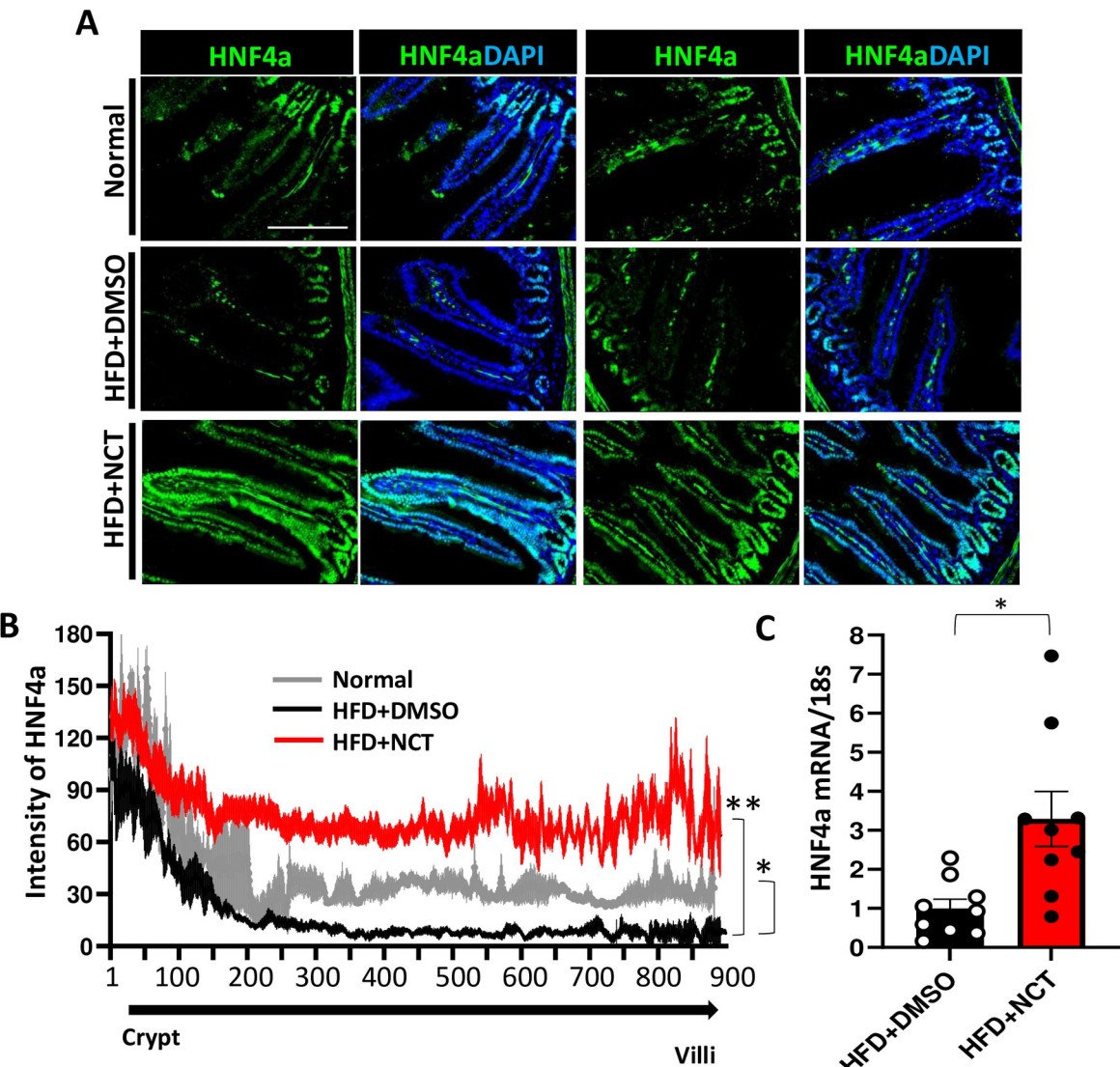

**Fig 1. NCT induces HNF4α expression in intestine.** 14week old DIO mice (C57BL/6J) fed HFD were injected IP with NCT (200mg/kg bid) [29] or DMSO for two weeks, followed by harvesting of intestines. **A**. HNF4α staining of a representative intestinal villus from intestine of each group. Frozen sections were stained with HNF4α (green) and DAPI (blue) in mice fed normal diet, HFD+DMSO or HFD+NCT. **B**. Quantification of HNF4α intensity along the length of the villus from crypt to villus tip in normal diet (gray), HFD +DMSO (black) and HFD+NCT (red) conditions. **C**. qPCR analysis of HNF4α mRNA in mouse small intestine (N = 9) normalized with 18s rRNA. Values represent the mean ± SEM of 3–7 mice. *$p < 0.05$, **$p < 0.01$ (HFD+DMSO vs Normal or HFD+NCT). Scale bar = 200μM.

identified the Paneth cell marker lysozyme as a node in a protein-protein interaction network that includes defensins (Fig 2), suggesting a strong involvement of HNF4α in Paneth cells.

## Genes important in inflammatory bowel disease were affected by NCT

Finding that NCT affected genes associated with obesity was expected, given that we were studying an obesity model, i.e., mice fed a high fat diet. However, finding many genes associated with IBD but not obesity was unexpected. The effect of NCT on those genes was verified by RT-PCR (Fig 3). A number of genes were found that are reduced in IBD were upregulated by NCT, including Slc10a2, Slc5a8, Trpm6, Enpp7, and Ddah1. Duox2, which was upregulated

**Table 2. IBD and obesity-associated genes.**

| Gene Name | Comment | Fold change +NCT/-NCT | REFERENCE | Associated Disease |
|---|---|---|---|---|
| Nos2 | Nitric oxide synthase. Plays a complex role in IBD | 83.2 | [34–36] | IBD |
| Mgat4c | Mannosyl (Alpha-1,3-)-Glycoprotein Beta-1,4-N-Acetylglucosaminyltransferase, Isozyme C (Putative). Induced in HT29 cells by macrophages. | 39.2 | [37, 38] | IBD |
| Slc10a2 | Sodium/bile acid cotransporter. Reduced in Crohn's. | 18.3 | [39–42] | IBD |
| Slc5a8 | Short chain fatty acid transporter. Decreased in UC. | 17.2 | [43] | IBD |
| Duox2 | Dual oxidase 2. Generates reactive oxygen species. Mutated in very early onset IBD patients. | 15.7 | [44] [45] [46] | IBD |
| Adcy8 | Adenyl cyclase 8. Upregulated in obesity. | 11 | [47] | obesity |
| Trpm6 | Intestinal absorption of magnesium. Reduced in IBD. | 10.8 | [48] | IBD |
| Enpp7 | Intestinal enzyme alkaline sphingomyelinase. Reduced in IBD. | 6.6 | [49] | IBD |
| Xpnpep2 | X-Prolyl Aminopeptidase 2. High expression in sites of inflammation in IBD and differential expression in colon and ileum in IBD. | 5.6 | [50] | IBD |
| Ddah1 | Dimethylarginine dimethylaminohydrolase 1. Regulates nitric oxide production. Downregulated in Crohn's disease. Genetically linked to obesity risk. | 5 | [51] | IBD obesity |
| Map3k6 | Mitogen-Activated Protein Kinase Kinase Kinase 6. Obesity-associated | 4.3 | [52] | obesity |
| Mylip | Myosin regulatory light chain interacting protein. Obesity-related gene | 4 | [53, 54] | obesity |
| Npl | N-Acetylneuraminate Pyruvate Lyase. Blood biomarker that differentiated patients with CD from those with UC and from noninflammatory diarrheal disorders | 4 | [55] | IBD |

Genes altered by NCT in the intestines of HFD+NCT-treated mouse intestine that are related to IBD and obesity. Fold change is HFD+DMSO vs HFD+NCT from GSE178435.

**Table 3. Paneth cell-related genes.**

| Gene Name | Comment | Fold change +NCT/-NCT | REFERENCE |
|---|---|---|---|
| Plb1 | Phospholipase B homolog. Expressed in Paneth cells | 367.8 | [58] |
| Defa22 | Paneth cell defensin. Reduced by IFNg | 122.5 | [59, 60] |
| Defa21 | Paneth cell defensin. | 110 | [59, 60] |
| Defa-rs1 | Paneth cell defensin | 20.5 | [61] |
| Defa5 | Paneth cell defensin. | 19.3 | [60] |
| Guca2a | Guanylate Cyclase Activator 2A. Endogenous activator of intestinal guanylate cyclase. Specific to Paneth cells. | 7.3 | [62, 63] |
| Ang4 | Angiogenin 4. Secreted by Paneth cells. | 6 | [60] [64] |
| Gm15292 | Defa40. Paneth cell defensin. | 5.1 | [61] |
| Mptx2 | mucosal pentraxin 2. Paneth cell marker. | 4.4 | [61] |
| Defa26 | defensin | 3.8 | [61] [59] |
| Igf2 | Insulin-like growth factor 2. Paneth cells absent in Igf2 KO | 3.8 | [65] |
| Mmp7 | Matrix metalloproteinase. Activates defensins in Paneth cells | 3.5 | [66] |
| C3 | Complement C3. Intracellular C3 activation upregulates Paneth cell activity | 3.3 | [67] |

Genes altered by NCT in the intestines of HFD+NCT-treated mouse intestine that are related to Paneth cells. Fold change is HFD+DMSO vs HFD+NCT from GSE178435.

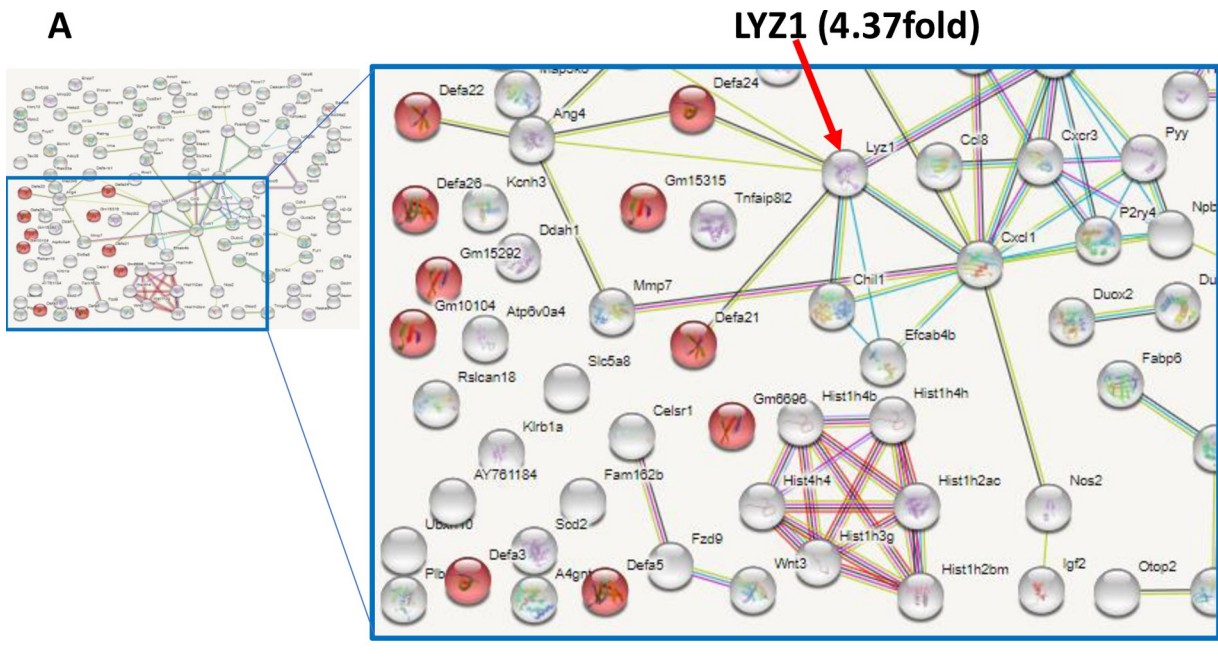

**Fig 2. STRING network and enrichment analysis identifying LYZ1 and defensins as candidates altered by NCT. A:** Diagram from STRING shows protein-protein interaction networks for 122 genes upregulated by >2.9 fold by NCT in HFD+NCT treated mouse intestine (GSE178435). Each node represents upregulated candidates and colored lines between the nodes indicate different types of evidence for protein-protein interactions as shown in the legend. Red nodes indicate protein domain, Defensin family (10 of 122 genes) and the red arrow indicates Lysozme1 (LYZ1) with fold change. **B:** STRING enrichment analysis for the top 122 upregulated gene candidates in NCT-treated mouse intestine, identifying defensins as enriched protein domains.

more than 15-fold in the RNA-seq dataset and about 3-fold in the RT-PCR confirmation (Fig 3E), is particularly interesting as it is causally implicated in IBD pathogenesis, being mutated in early onset Crohn's disease. Interestingly, the IBD-associated genes altered by NCT were not affected by HFD with the exception of Slc10a2, a bile acid transporter that is reduced in Crohn's disease (Fig 3, Table 2).

## Paneth cells were decreased by HFD, which was strongly reversed by NCT

A large number of the genes affected by NCT are expressed in Paneth cells, including a number of defensins, which are the hallmarks of Paneth cells (Table 3). To pursue that finding, which was unexpected since Paneth cells have not been studied extensively in obesity or as downstream targets of HNF4α [56, 57]. To pursue the effect of NCT on Paneth cells, we

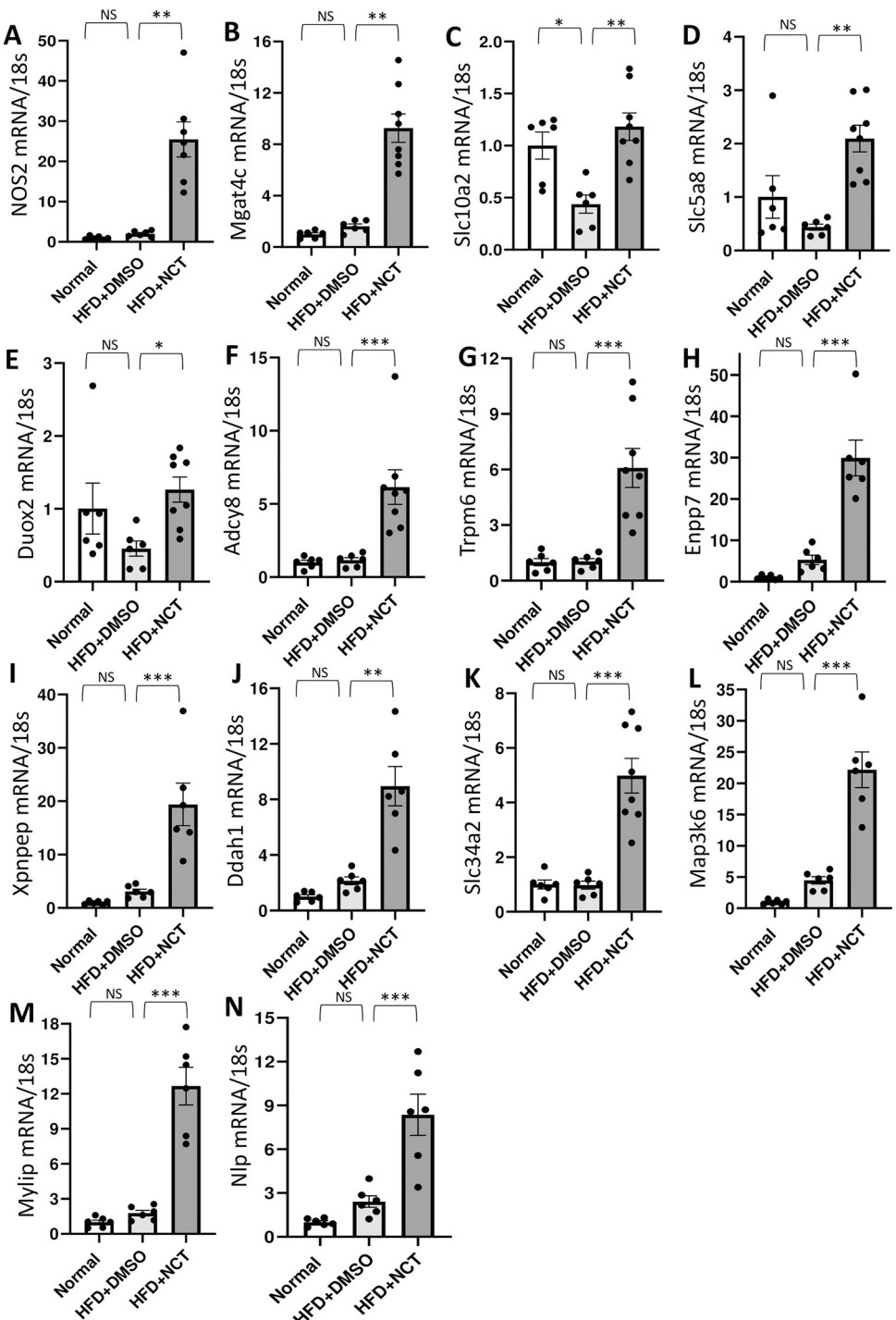

**Fig 3. QPCR confirmation of IBD and obesity-associated genes induced by NCT. A-N:** qPCR analysis in mouse small intestine of Nos2, Mgat4c, Slc10a2, Slc5a8, Duox2, Adcy8, Trpm6, Enpp7, Xpnpep, Ddah1, Slc34a2, Map3k6, Mylip, Nlp mRNA expression normalized with 18s rRNA (Normal chow, N = 6, HFD+DMSO, N = 6 and HFD+NCT, N = 6–8). Dots indicate individual mice. Values represent the mean ± SEM. NS = non-significant, $^*p<0.05$, $^{**}p<0.01$, $^{***}p<0.001$.

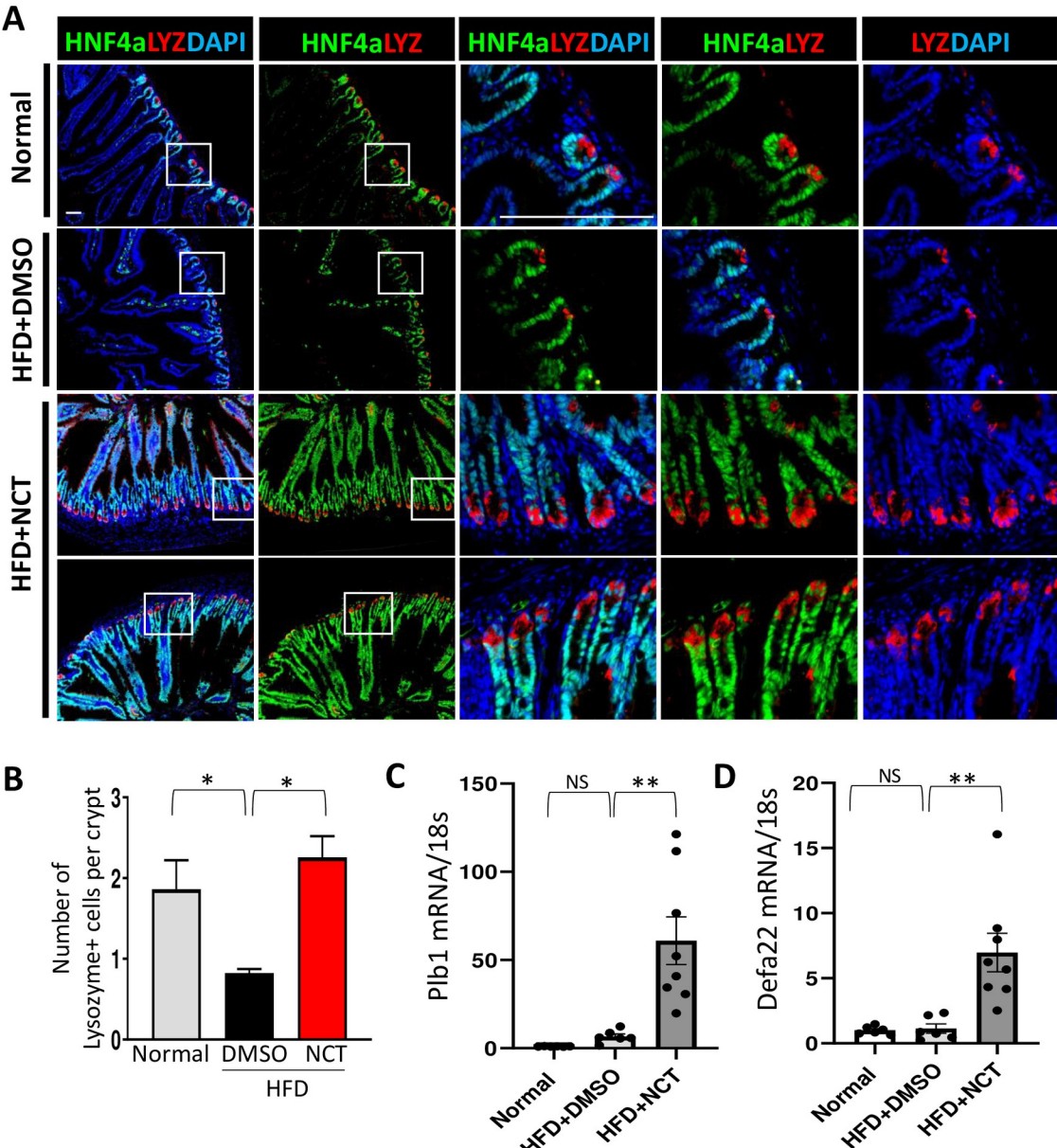

**Fig 4. NCT induced recovery of Paneth cells in intestine. A**. Frozen sections of intestine from the mice described in Fig 1 were stained with HNF4α (green), Lysozyme (red) and DAPI (blue) in mice fed normal diet, HFD+DMSO or HFD+NCT. White box indicates high power view in adjacent panel. **B**. Quantification of the number of lysozyme-positive cells per intestinal crypt (HFD+DMSO vs. Normal or HFD+NCT). **C, D**. qPCR analysis in mouse small intestine of Plb1 and Defa22 mRNA expression normalized with 18s rRNA (N = 6–8). NS = non-significant, $^*p<0.05$, $^{**}p<0.01$. Values represent the mean ± SE of 3–7 mice. Scale bar = 200μM.

performed immunostaining for the Paneth cell marker lysozyme. There were decreased numbers of lysozyme-expressing Paneth cells in the intestines of DIO mice (Fig 4A and 4B). NCT almost completely reversed that effect, consistent with the RNA-seq data (Fig 4A and 4B), which was confirmed by RT-PCR analysis of Plb1 and Defa22 expression (Fig 4C and 4D). There was no effect of HFD of NCT on the morphology of the intestinal epithelium, supporting a model in which the repression of HNF4α by HFD and its activation by NCT affected

Paneth cell gene expression rather than the actual number of Paneth cells, consistent with findings in the HNF4α intestinal cell knockout mouse [57].

## Discussion

The principal finding reported here is that in a mouse model of diet-induced obesity the genes induced by NCT, a potent HNF4α agonist, were highly relevant to IBD. There is a complex relationship between obesity, fatty acids, and IBD [68], and the data presented here suggest that HNF4α may be important in their interrelationship. However, it is important to note that most of the IBD-relevant genes affected by NCT were not altered by HFD.

The intestines used in this study came from an experiment designed to study the role of NCT in diet-induced obesity. Focusing on obesity was logical, as we had shown previously that fatty acids are HNF4α antagonists [18]. Thus, we hypothesized that an agonist would have therapeutic benefit, which was borne out by the dramatic reduction in hepatic steatosis induced by NCT [29]. As expected, some obesity-associated genes were affected by NCT.

Surprisingly, the most prominent class of genes induced by NCT were those expressed in Paneth cells. Paneth cells markers were reduced in the HNF4α intestinal cell knockout mouse but this was not understood to be a direct effect of HNF4α [57]. The finding here that Paneth cell gene expression is strongly enhanced by NCT demonstrates an important role for HNF4α in that cell type. While HNF4a had a strong effect on Paneth cells, that did not appear to be due to direct effects on Paneth cell genes, as a study of ChIP-seq in the intestine did not identify genes expressed in Paneth cells [11]. Rather, genes expressed in the brush border epithelium appeared to be directly downstream of HNF4a. Those genes included three IBD-associated genes that were upregulated by NCT: Slc10a2, Trpm6, and Enpp7 (Table 2).

While Paneth cells are decreased in the intestines of mice fed HFD [56], they have not been thought to play a major role in obesity. However, they have been shown to play a central role in IBD [69–71]. They play key roles in intestinal barrier function and regulation of the intestinal microbiome [72]. Thus, the strong upregulation of genes important in Paneth cell function bodes well for the therapeutic potential of HNF4α agonists in IBD.

Not only did NCT reverse the reduction of HNF4α that occurred in the intestine of DIO mice; it actually increased HNF4α expression to a level higher than in mice fed a normal chow diet. HNF4α functions in transcriptional feedback loops to control its own promoter [18, 23, 73–75], making its downregulation correctable by an HNF4α agonist as we have observed multiple times [23, 29], including here. It is interesting to note that TNFα, a cytokine that plays a central role in IBD pathogenesis [76] and is the target of IBD drugs [77], inhibits HNF4α activity through a pathway involving NFkB [78].

HNF4α is believed to exist in the active state at baseline [2]. However, it is not known whether it is maximally active. The finding that NCT is able to increase HNF4α expression, which is itself an HNF4α target [18, 23, 29, 73–75], to a level greater than baseline (Fig 1) indicates that it is likely that the level of HNF4α activity at baseline is less than maximal. This could be because of some degree of occupancy of the HNF4α ligand binding pocket by fatty acids, which repress HNF4α activity [18]. NCT could be increasing the percentage of HNF4α molecules that are in the activated state by increasing the percentage of time that HNF4α has NCT versus a fatty acid in its LBP, leading to higher level of downstream gene expression, including *HNF4α* itself [29].

HNF4α is generally considered to be a transcriptional activator and so would generally be expected to increase the level of downstream gene expression. This was the case with our data, where genes such as Slc10a2, Slc5a8, and Ddah1that are reduced in IBD were increased by NCT (Table 2). Genes expressed in Paneth cells were also upregulated by NCT (Table 3). The

upregulation by NCT of multiple genes downregulated in IBD and affected by high fat diet in Paneth cells demonstrates a role for HNF4α in important aspects of IBD and suggests that HNF4α agonists may be good candidates as IBD therapeutics.

## Acknowledgments

We thank the following SBP Core Facilities: Imaging, Animal, Genomics, Conrad Prebys Center for Chemical Genomics, Histology.

## Author Contributions

**Conceptualization:** Fred Levine.

**Funding acquisition:** Fred Levine.

**Investigation:** Seung-Hee Lee, Vimal Veeriah.

**Methodology:** Seung-Hee Lee.

**Project administration:** Fred Levine.

**Supervision:** Fred Levine.

**Writing – original draft:** Seung-Hee Lee, Fred Levine.

**Writing – review & editing:** Seung-Hee Lee, Vimal Veeriah, Fred Levine.

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
