## [Decision Letter · Decision Letter 0]

27 Oct 2021

PONE-D-21-30275A potent HNF4α agonist reveals that HNF4α controls genes important in inflammatory bowel disease and Paneth cellsPLOS ONE

Dear Dr. Levine,

Thank you for submitting your manuscript to PLOS ONE. After careful consideration, we feel that it has merit but does not fully meet PLOS ONE’s publication criteria as it currently stands. Therefore, we invite you to submit a revised version of the manuscript that addresses the points raised during the review process. Specifically, it was felt that the authors need to perform additional studies to show whether the effects of HNF4 are direct or indirect.  The suggested additional studies including western analysis and CHIP assays may further strengthen the impact of the manuscript.

We look forward to receiving your revised manuscript.

Kind regards,

Pradeep Dudeja

Academic Editor

PLOS ONE

Journal Requirements:

2. To comply with PLOS ONE submissions requirements, in your Methods section, please provide additional information on the animal research and ensure you have included details on (1) number of animals used in the study, and (2) basic housing.

[We thank the following SBP Core Facilities: Imaging, Animal, Genomics, Conrad Prebys Center for Chemical Genomics, Histology. Sources of Support:  This work was funded by the Sanford Children’s Health Research Center and Brightseed, Inc. FL holds equity in Brightseed.]

 [This work was funded by the Sanford Children’s Health Research Center and Brightseed, Inc. The funders had no role in study design, data collection and analysis, decision to publish, or preparation of the manuscript.]

[I have read the journal's policy and the authors of this manuscript have the following competing interests: FL holds equity in Brightseed.] 

Reviewers' comments:

Reviewer's Responses to Questions

**Comments to the Author**

1. Is the manuscript technically sound, and do the data support the conclusions?

Reviewer #1: Yes

Reviewer #2: Yes

2. Has the statistical analysis been performed appropriately and rigorously? 

Reviewer #1: Yes

Reviewer #2: I Don't Know

3. Have the authors made all data underlying the findings in their manuscript fully available?

Reviewer #1: Yes

Reviewer #2: Yes

4. Is the manuscript presented in an intelligible fashion and written in standard English?

Reviewer #1: Yes

Reviewer #2: Yes

5. Review Comments to the Author

Reviewer #1: In this interesting study, the authors have investigated the therapeutic effect of the potent HNF4� agonist, N-trans caffeoyltyramine (NCT) on gene expression in the intestine. This study is based on previous findings that demonstrated the therapeutic potential of the novel HNF4� activator, NCT in the liver by reversing severe NAFLD symptoms (hepatic steatosis) in diet-induced obese mice. For the current study, vehicle (DMSO) or NCT was administered to high fat diet mice for 14 days (2 weeks) followed by RNAseq, RT-PCR, and Immunofluorescence analysis. The results showed that NCT upregulated genes in the small intestine that play important role in IBD and that are shown to be downregulated in IBD. Moreover, NCT also reversed the decrease in lysozyme expression (Paneth cell marker) in the small intestine in high fat diet fed mice. Impaired Paneth cell function and intestinal barrier integrity have shown to be the hallmarks of IBD. Therefore, these data suggest that HNF4� could be a therapeutic target for IBD and that NCT could act as a candidate therapeutic in treating IBD. The manuscript for the most part is easily understandable, straight-forward and the results are clearly presented. The abstract adequately summarizes the data and is concise. The strength of the manuscript includes the therapeutic relevance of the study with respect to the important role of HNF4� agonist, NCT in the upregulation of genes that are decreased in IBD while the weaknesses relate to the lack of utilizing an established mouse model of intestinal disease (colitis) to examine the beneficial effect of NCT.

1. Since most of the IBD associated genes upregulated by NCT are independent of HFD (high fat diet) effects (Fig. 3E), then it must be assumed that NCT will also show a similar phenomenon in mice fed with normal chow diet. The authors should elaborate and discuss more on this aspect.

2. Although, the immunofluorescence (IF) images are convincing (Fig. 1A), it will be important to corroborate IF results with Western blot utilizing small intestinal tissue lysates and specific HNF4� antibody.

3. In addition to the lysozyme expression (Paneth cell marker), the authors should also show the altered expression of certain Paneth cell defensins in HFD mice with or without NCT by qRT-PCR.

4. It is not clear which region of the small intestine (jejunum vs ileum) was used for the studies?

Reviewer #2: Study by Seung- Hee Lee et al- Entitled “A potent HNF4α agonist reveals that HNF4α controls genes important in inflammatory bowel disease and Paneth cells” has interesting findings. Although, there are some concerns that need to be addressed to strengthen the manuscript.

Major concerns:

1. In this study, the authors have showed that the decreased expression of the IBD associated gene and Paneth cell markers by HFD and their reversal by NCT. However, the data does not suggest that the effects are directly due to HNF4A, so it will be important to examine the activation of HNF4A by NCT.

2. ChIP assays or other studies should be provided to prove that NCT increases the binding of HNF4A to the promoter of its target gene.

3. Authors should also examine or provide detail that NCT is a specific agonist only to HNF4A, and it is not producing effects indirectly.

Minor concerns:

1. Authors should rephrase the abstract for better grammar and flow.

2. Detail of RT-PCR primers and sting analysis should be provided in the methods and materials section.

3. Authors should include the previous study done regarding regulation of stem cells in HNF4A KO mice

4. Authors should also include a column in table 2 for associated diseases same as table 1.

6. PLOS authors have the option to publish the peer review history of their article (what does this mean?). If published, this will include your full peer review and any attached files.

Reviewer #1: No

Reviewer #2: No

---

## [Author Response · Author response to Decision Letter 0]

17 Nov 2021

Reviewer #1: In this interesting study, the authors have investigated the therapeutic effect of the potent HNF4� agonist, N-trans caffeoyltyramine (NCT) on gene expression in the intestine. This study is based on previous findings that demonstrated the therapeutic potential of the novel HNF4� activator, NCT in the liver by reversing severe NAFLD symptoms (hepatic steatosis) in diet-induced obese mice. For the current study, vehicle (DMSO) or NCT was administered to high fat diet mice for 14 days (2 weeks) followed by RNAseq, RT-PCR, and Immunofluorescence analysis. The results showed that NCT upregulated genes in the small intestine that play important role in IBD and that are shown to be downregulated in IBD. Moreover, NCT also reversed the decrease in lysozyme expression (Paneth cell marker) in the small intestine in high fat diet fed mice. Impaired Paneth cell function and intestinal barrier integrity have shown to be the hallmarks of IBD. Therefore, these data suggest that HNF4� could be a therapeutic target for IBD and that NCT could act as a candidate therapeutic in treating IBD. The manuscript for the most part is easily understandable, straight-forward and the results are clearly presented. The abstract adequately summarizes the data and is concise. The strength of the manuscript includes the therapeutic relevance of the study with respect to the important role of HNF4� agonist, NCT in the upregulation of genes that are decreased in IBD while the weaknesses relate to the lack of utilizing an established mouse model of intestinal disease (colitis) to examine the beneficial effect of NCT.

CRITIQUE 1. Since most of the IBD associated genes upregulated by NCT are independent of HFD (high fat diet) effects (Fig. 3E), then it must be assumed that NCT will also show a similar phenomenon in mice fed with normal chow diet. The authors should elaborate and discuss more on this aspect.

RESPONSE: The reviewer is correct that most of the IBD-associated genes were not affected by HFD. We agree that this point is worthy of greater emphasis and so have modified the manuscript to make this point more clearly (Lines 210-212, 255-256). 

CRITIQUE 2. Although, the immunofluorescence (IF) images are convincing (Fig. 1A), it will be important to corroborate IF results with Western blot utilizing small intestinal tissue lysates and specific HNF4� antibody.

RESPONSE: To address the reviewer’s request for additional data on HNF4a expression, we performed RT-PCR analysis for HNF4a mRNA, finding that there was a significant increase induced by NCT. That has been added as Figure 1C. 

CRITIQUE 3. In addition to the lysozyme expression (Paneth cell marker), the authors should also show the altered expression of certain Paneth cell defensins in HFD mice with or without NCT by qRT-PCR.

RESPONSE. To address the concern of the reviewer, we have added additional data (Figure 4C, D). 

CRITIQUE 4. It is not clear which region of the small intestine (jejunum vs ileum) was used for the studies?

RESPONSE. Ten cm of intestine proximal to the duodenum, and so consisting primarily of jejunum, was removed. This information has been added to the Materials and Methods- section on Mice. 

Reviewer #2: Study by Seung- Hee Lee et al- Entitled “A potent HNF4α agonist reveals that HNF4α controls genes important in inflammatory bowel disease and Paneth cells” has interesting findings. Although, there are some concerns that need to be addressed to strengthen the manuscript.

Major concerns:

CRITIQUE 1. In this study, the authors have showed that the decreased expression of the IBD associated gene and Paneth cell markers by HFD and their reversal by NCT. However, the data does not suggest that the effects are directly due to HNF4A, so it will be important to examine the activation of HNF4A by NCT.

RESPONSE. The issue of whether NCT interacts directly with HNF4a is an important question. Our previous work with NCT examined the issue of direct interaction and we have modified the manuscript to make this important point requested by the reviewer (Lines (79-81). 

CRITIQUE 2. ChIP assays or other studies should be provided to prove that NCT increases the binding of HNF4A to the promoter of its target gene.

RESPONSE. The reviewer requests ChIP assays to determine whether the effects of NCT on downstream genes are direct or indirect. Fortunately, a comprehensive ChIP-Seq analysis of the role of HNF4 in the intestine was published recently: Chen L, et al. The nuclear receptor HNF4 drives a brush border gene program conserved across murine intestine, kidney, and embryonic yolk sac. Nat Commun. 2021 May 17;12(1):2886. doi: 10.1038/s41467-021-22761-5. PMID: 34001900; PMCID: PMC8129143. They did not find that the Paneth cell genes that we studied to be direct targets of HNF4a, but a few of the genes associated with IBD were identified as direct targets of HNF4a. The manuscript has been modified to discuss more extensively the results from this paper along with additional discussion of the important issue identified by the reviewer about direct versus indirect effects of HNF4a (Lines 266-270). 

CRITIQUE 3. Authors should also examine or provide detail that NCT is a specific agonist only to HNF4A, and it is not producing effects indirectly.

RESPONSE. As pointed out by the reviewer, the specificity of small molecule compounds is always a concern. We have examined this issue in our previous paper on NCT using siRNA to HNF4a, finding that it ablated the effect of NCT in cultured cells (Lines (79-81). 

Minor concerns:

1. Authors should rephrase the abstract for better grammar and flow.

RESPONSE. The abstract has been edited as requested. 

2. Detail of RT-PCR primers and sting analysis should be provided in the methods and materials section.

RESPONSE. This information has been added as Table 1. 

3. Authors should include the previous study done regarding regulation of stem cells in HNF4A KO mice

RESPONSE. We have added a citation to a recent paper on the role of HNF4a in intestinal stem cell renewal (Chen L, et al. HNF4 Regulates Fatty Acid Oxidation and Is Required for Renewal of Intestinal Stem Cells in Mice. Gastroenterology. 2020 Mar;158(4):985-999.e9. doi: 10.1053/j.gastro.2019.11.031. Epub 2019 Nov 22. PMID: 31759926; PMCID: PMC7062567) and hope that this is the citation requested by the reviewer.

4. Authors should also include a column in table 2 for associated diseases same as table 1.

RESPONSE. We did not include a column for associated diseases in table 2, since that table included only genes expressed in Paneth cells and the point of the table is to emphasize that many genes expressed in Paneth cells were affected by NCT. To the best of our knowledge, the only disease that has been well studied with regard to Paneth cells is IBD, an association that is discussed at length in the manuscript.

---

## [Decision Letter · Decision Letter 1]

14 Mar 2022

A potent HNF4α agonist reveals that HNF4α controls genes important in inflammatory bowel disease and Paneth cells

PONE-D-21-30275R1

Dear Dr. Levine,

We’re pleased to inform you that your manuscript has been judged scientifically suitable for publication and will be formally accepted for publication once it meets all outstanding technical requirements.

Kind regards,

Pradeep Dudeja

Academic Editor

PLOS ONE

Additional Editor Comments (optional):

Reviewers' comments:

Reviewer's Responses to Questions

**Comments to the Author**

1. If the authors have adequately addressed your comments raised in a previous round of review and you feel that this manuscript is now acceptable for publication, you may indicate that here to bypass the “Comments to the Author” section, enter your conflict of interest statement in the “Confidential to Editor” section, and submit your "Accept" recommendation.

Reviewer #1: All comments have been addressed

Reviewer #2: All comments have been addressed

2. Is the manuscript technically sound, and do the data support the conclusions?

Reviewer #1: Yes

Reviewer #2: Yes

3. Has the statistical analysis been performed appropriately and rigorously? 

Reviewer #1: Yes

Reviewer #2: Yes

4. Have the authors made all data underlying the findings in their manuscript fully available?

Reviewer #1: Yes

Reviewer #2: Yes

5. Is the manuscript presented in an intelligible fashion and written in standard English?

Reviewer #1: Yes

Reviewer #2: Yes

6. Review Comments to the Author

Reviewer #1: The authors have duly addressed all the concerns raised by the reviewer. The revised version of the manuscript is now ready to be accepted

Reviewer #2: (No Response)

7. PLOS authors have the option to publish the peer review history of their article (what does this mean?). If published, this will include your full peer review and any attached files.

Reviewer #1: **Yes: **Seema Saksena

Reviewer #2: No

---

## [Editor Report · Acceptance letter]

17 Mar 2022

PONE-D-21-30275R1 

A potent HNFα agonist reveals that HNF4α controls genes important in inflammatory bowel disease and Paneth cells 

Dear Dr. Levine:

I'm pleased to inform you that your manuscript has been deemed suitable for publication in PLOS ONE. Congratulations! Your manuscript is now with our production department. 

Kind regards, 

on behalf of

Dr. Pradeep Dudeja 

Academic Editor

PLOS ONE